# Beyond Information Provision: Analysis of the Roles of Structure and Agency in COVID-19 Vaccine Confidence in Ethnic Minority Communities

**DOI:** 10.3390/ijerph20217008

**Published:** 2023-11-01

**Authors:** Shoba Poduval, Atiya Kamal, Sam Martin, Amin Islam, Chandrika Kaviraj, Paramjit Gill

**Affiliations:** 1UCL Research Department of Primary Care & Population Health, Royal Free Hospital, Rowland Hill Street, London NW3 2PF, UK; 2School of Social Sciences, Birmingham City University, 4 Cardigan Street, Birmingham B4 7BD, UK; atiya.kamal@bcu.ac.uk; 3Rapid Research Evaluation and Appraisal Lab (RREAL), Department of Targeted Intervention, University College London (UCL), Charles Bell House, 43–45 Foley Street, London W1W 7TY, UK; sam.martin@ucl.ac.uk; 4Vaccines and Society Unit, Oxford Vaccine Group, Oxford University, Oxford OX3 7LE, UK; 5Patient and Public Involvement Authors, UCL Research Department of Primary Care & Population Health, Royal Free Hospital, Rowland Hill Street, London NW3 2PF, UK; 6Warwick Medical School, University of Warwick, Coventry CV4 7AL, UK; p.gill.1@warwick.ac.uk

**Keywords:** ethnic and racial minorities, ethnicity, COVID-19, COVID-19 vaccines, vaccination hesitancy, health inequities, primary health care, public health, health promotion, systemic racism

## Abstract

People from Black and Asian backgrounds are more likely to die from COVID-19 but less likely to be vaccinated, threatening to exacerbate health inequalities already experienced by ethnic minority groups. The literature suggests that mistrust rooted in structural inequality (including socioeconomic position and experience of racism) may be a key barrier to COVID-19 vaccine uptake. Understanding and addressing structural inequality is likely to lead to longer-term impacts than information alone. The aim of this study is to draw on health and sociological theories of structure and agency to inform our understanding of how structural factors influence vaccine confidence. We conducted qualitative interviews and focus groups with 22 people from London and the surrounding areas from December 2021 to March 2022. Fourteen participants were members of the public from ethnic minority backgrounds, and seven were professionals working with the public to address concerns and encourage vaccine uptake. Our findings suggest that people from ethnic minority backgrounds make decisions regarding COVID-19 vaccination based on a combination of how they experience external social structures (including lack of credibility and clarity from political authority, neglect by health services, and structural racism) and internal processes (weighing up COVID-19 vaccine harms and benefits and concerns about vaccine development and deployment). We may be able to support knowledge accumulation through the provision of reliable and accessible information, particularly through primary and community care, but we recommend a number of changes to research, policy and practice that address structural inequalities. These include working with communities to improve ethnicity data collection, increasing funding allocation to health conditions where ethnic minority communities experience poorer outcomes, greater transparency and public engagement in the vaccine development process, and culturally adapted research recruitment processes.

## 1. Introduction

The United Kingdom (UK) COVID-19 vaccine campaign started in December 2020 with primary care services (including general practitioner (GP) practices and community pharmacies) leading vaccine delivery [1]. The most recently released data on vaccination rates in adults shows that by March 2023, 75.8% of the general population had received at least three COVID-19 vaccinations [2]. Whilst vaccine delivery has been successful in reaching a large number of people, vaccine uptake has been particularly low in ethnic minority groups (with 34.8% of Pakistani adults, 36.3% of Bangladeshi adults and 41.8% of Black African adults continuing to a fourth vaccination compared to 78.1% of White adults) [2]. Now that we have moved into COVID-19 recovery, the government, healthcare professionals and communities face the challenge of building vaccine confidence and addressing health inequalities faced by ethnic minority groups.

Vaccine hesitancy has been defined by the World Health Organization (WHO) as a refusal or delay in vaccine acceptance [3] and is not a new concept. In 2014, the WHO SAGE working group developed the ‘3 C’s’ model to categorise reasons for hesitancy into three areas: Convenience (access), Confidence (trust), and Complacency (perceived risk) [4]. Whilst vaccine hesitancy has been conceptualised in the general population, less is known about vaccine hesitancy in people from ethnic minority backgrounds. During the COVID-19 pandemic, a rapid assessment of the underlying factors leading to vaccine hesitancy in ethnic minority groups was undertaken, and the data demonstrated that the factors associated with vaccine hesitancy were younger age, less education, lower gross annual income and living in areas of high deprivation [5,6]. Research since the pandemic, including a semi-structured interview study by Woodhead et al. [7], has further emphasised the role of wider inequalities in vaccine mistrust, including socioeconomic disadvantage and exposure to and anticipation of discrimination [8,9]. This supports findings from research on ethnic health inequalities by medical sociologists, including Harding, Maxwell and Nazroo, who suggest that we need to better understand the role of structural factors (including socioeconomic position and experience of racism) in order to unpack the processes through which ethnic minority status leads to inequalities in health [10]. Understanding and addressing these processes is likely to lead to longer-term impacts to reduce vaccine inequalities than information alone. This is supported by the evidence on parental vaccination decisions which suggests that interventions focused on vaccine knowledge, education and messaging change vaccine attitudes and increase vaccination rates in the short-term. However, in the longer-term, interventions solely focused on vaccine messaging may lead to dissatisfaction due to deeper-rooted beliefs driven by individual experiences and beliefs resulting from structural inequalities [11,12,13,14].

### 1.1. Theoretical Underpinnings

We will draw on structuration theory [15] and critical realist social theory [16,17] to inform our understanding of how the social environment (‘social structures’) interacts with human action (‘agency’) to influence vaccine acceptance by ethnic minority communities. The theory of structuration groups social structures into ‘external’ and ‘internal’ [15,18,19]. External structures represent the social structures in which people operate and are mediated by institutional infrastructures and power relations, political and medical ‘authority’, social position and the associated identity and network of social interactions, and race, racial inequality and racialized social environments where Black and other non-White group members are routinely stigmatised [20]. Structural constraints and opportunities are created by health and social policies, and this impacts people’s abilities to leverage resources and be active in their own health [21]. Internal structures (or processes) refer to how and what people ‘know’ and can be analytically divided into ‘general dispositions’ (world-views, morals, principles and attitudes) and ‘conjuncturally-specific knowledge’ (knowledge specific to the immediate decision-making situation) [15,18,19]. The action (active agency) produced can have intended and unintended impacts on internal and external structures, which may be reproduced or changed [15,22].

Within critical realist social theory, the social world is a layered, complex and open system consisting of people (agents) actively and reflexively drawing upon both their attitudes and knowledge (internal structures) and social interactions (external structures) to generate behaviours [22,23,24]. People belong to and are influenced by multiple institutions and structural relations, which can facilitate the sharing of common concerns and make people reflect and act differently [17]. The use of critical realist social theory allowed us to identify the complex interactions between people and social structures and a possible causal pathway for how this may influence COVID-19 vaccine confidence. This is presented in a summary framework.

### 1.2. Aim and Objectives

The aim of this research study is to improve the understanding of COVID-19 vaccine confidence and the structural factors influencing confidence in low-uptake ethnic minority groups in London and the surrounding areas.

The specific objectives are to:Explore which structural factors influence decision-making, including external structures (education, employment, income, social identity, positioning and networks and experiences of structural racism) and internal structures (morals, values, attitudes and knowledge);Explore actions that would enable primary care and public health campaigns to improve confidence in COVID-19 vaccination in ethnic minority groups.

## 2. Materials and Methods

### 2.1. Study Design

This was a qualitative study consisting of focus groups and interviews with people from ethnic minority (non-White British) communities. Both focus groups and interviews were offered. Focus groups are particularly useful for gaining insights from minority ethnic groups because they help gain an understanding of group “norms”, values and “community” responses [25,26,27,28]. Two groups took part in the focus groups (members of the public only), and the remaining participants expressed a preference to be interviewed one-on-one.

### 2.2. Study Setting

The interviews and focus groups were conducted online in London, UK, from December 2021 to March 2022. Vaccine rollout in the UK started in December 2020. COVID-19 vaccine uptake data showed that the four ethnic minority groups with the lowest uptake were Bangladeshi, African, Caribbean and Pakistani [29,30]. Data also showed that London was the lowest uptake area in England (64% of people aged over 16 years old had had at least one dose of the vaccine compared to an average of 89% in England) [30].

At the start of the study in December 2021, new variants of the COVID-19 virus were emerging, and hospitalisations were increasing, particularly in Bangladeshi and Pakistani communities. A booster vaccine was being offered with the aim of achieving a high uptake by January 2022. There was also discussion about mandatory vaccination for NHS and social care workers.

### 2.3. Patient and Public Involvement (PPI)

SP led a participatory community engagement project exploring under-representation in health research by ethnic minority communities (“Diverse Voices”) in June 2021, which led to the formation of the PPI group supporting this study (CKand AI) [31]. The PPI representatives were part of the core research team and advised on the study design, recruitment strategy, research flyer, topic guide and interpretation of findings. The approach to PPI was based on NIHR guidance on inclusive public involvement [32,33].

### 2.4. Sampling and Recruitment

All ethnic groups apart from White British with current or previous concerns about COVID-19 vaccines, leading to delay or refusal to be vaccinated, were eligible to take part in the study. Interpreting services for Bengali and Urdu speakers were offered but not used. Access and skills to use MS Teams or a telephone were needed. Eligible participants were identified and recruited through established Patient and Public Involvement (PPI) networks, community organisations and snowball sampling. Organisations in areas with larger Bangladeshi, Pakistani, African and Caribbean populations (including Newham, Tower Hamlets and Hackney) were contacted by email with the research flyer and asked to disseminate it via email, newsletters and WhatsApp. Professionals were eligible if they worked in the NHS, local authorities or voluntary and community organisations (VCSOs) providing COVID-19 vaccine information and access to ethnic minority communities in London. Professionals were identified from community outreach events and snowball sampling and were contacted by email with participant and consent information.

Ethical approval for this study was received from the University College London Research Ethics Committee (ID no 6761/002).

### 2.5. Data Collection and Analysis

Interviews were conducted by lead author SP, who is experienced in qualitative methods and working with ethnic minority communities. A topic guide (Appendix A) was developed by SP, informed by the literature, PPI representatives and Co-Is. The topic guide covered the following areas: social background; experiences of the pandemic and offer of vaccination; access to vaccine information; factors influencing decision-making; comparison with other vaccines; and suggestions for improving COVID-19 vaccine information and access. Focus groups and interviews were conducted online and recorded using MS Teams to comply with the guidance on minimising the spread of COVID-19. The focus groups lasted approximately 60–90 min, and interviews ranged from 30–60 min. Participants were given a shopping voucher as a thank-you for their time. All participants had sufficient English language skills to conduct the focus groups and interviews in English. Professional transcription services were used to transcribe the recordings verbatim. SP anonymised the transcripts and checked the transcripts against the recordings for accuracy.

The transcripts were managed using NVivo Version 12 software (Lumivero, Denver, United States) [34]. Inductive reflective thematic analysis, informed by Braun and Clarke was used to analyse data in the ‘first-level analysis’ (working from the data up) [35]. This approach ensured that analytic categories were obtained gradually from the data and experiences of the participants themselves [36]. Data analysis took place alongside the data collection so that early concepts were identified, and the topic guide was updated accordingly to explore key issues in more detail. Led by SP, the process started with data familiarisation, followed by data coding when each phrase, line and paragraph of the transcripts was scrutinised in detail. Ideas and concepts were labelled with a code and the codes were sorted so that data with similar content were located together in NVivo [37]. Codes from a sample of transcripts were shared and discussed at two researchers’ meetings with SP and AK, a qualitative researcher with significant expertise in vaccine hesitancy. The analysis was also informed by a participatory action workshop to ensure the findings were grounded in the experiences of the participants and people affected by the research question [38]. An online 90 min participatory workshop was held with a small group (*n* = 4) of PPI representatives and participants who responded to an email invitation to take part. Two anonymised transcripts were shared with the group in advance, and co-production methods [39,40] were used to discuss and share ideas about the meaning emerging from sections of the transcripts. This included individuals documenting their ideas collectively by using a digital whiteboard (Jamboard (Google, Santa Clara, United States) [41]).

Once the codes had been shared and validated in the data meetings and workshop, they were grouped into initial themes by SP and shared with all co-authors. To go beyond a description of the data and produce a more exploratory analysis, we produced a summary of the main themes and sub-themes from the first-level thematic analysis, followed by an overall thematic framework examining the connections between themes and existing structuration and critical realist social theory [37,42].

## 3. Results

### 3.1. Participants

Focus groups and interviews were conducted with fourteen members of the public, and eight interviews were conducted with professionals involved in COVID-19 vaccination campaigns (a total of *N* = 22). *N* = 4 of the professionals were male, and *n* = 4 were female. A total of 6/8 (75%) professionals were from ethnic minority backgrounds. Of the members of the public, the age of the participants ranged from 21–67 years old. *N* = 9 were female, and *n* = 5 were male. A total of 10/14 (71.4%) participants were from African, Caribbean, Pakistani or Bangladeshi backgrounds. The remaining participants were mixed race (Caribbean/White British), Indian or preferred not to say. A total of 13/14 (92.8%) participants were in employment (4/14 or 28.6% in the NHS). The demographic characteristics of members of the public are summarised in Table 1.

### 3.2. First-Level Analysis

The following overarching themes were identified:Lack of information credibility and clarity;Discrimination and barriers to healthcare;Personal beliefs, characteristics and experiences influencing vaccine choice;Social networks and community cohesion.

#### 3.2.1. Lack of Information Credibility and Clarity

Some participants lacked trust in official information and the government. This included trust in data on the number of deaths in different ethnic groups. Participants described how they did not share their ethnicity when asked and their concern that if others did the same, this would result in inaccurate data.


*“I find that it is a bit misleading. I don’t know, I haven’t seen enough reports that make me feel encouraged by the data that’s been given. As I said, when they ask me my own ethnicity, I always say unknown, I would prefer not to say, but for me the figures are not correct anyway. They’re not really reflecting the fact that there may be other people who don’t put down their ethnicity or race, they just don’t do that. So I don’t think the figures are really a true reflection of what is happening.”*
(P01, ethnicity not disclosed, 57 years, female). 

Some participants expressed that they did trust the government and rejected the idea that ethnic minority communities were not supported adequately.


*“I trust the government information that they give. I don’t think there is any difference between member of the—of any community. If people, they want to pick up on this, saying that, you know, the government didn’t’ do enough for BME (Black and minority ethnic people) or whatever, I don’t think it’s true”.*
(P05, Pakistani, 36 years, female). 

Other participants described how inconsistency in the government’s messaging led to contradiction and confusion, which undermined their trust in politicians.


*“Compared to other countries our government has been very inconsistent. One example I think I do have is like maybe New Zealand. And I think it was a Scandinavian country, they were very clear, they were very like, “Well, this is how we’re doing it and this is why we’re doing it.” And they spoke directly to the people. In the UK, it’s the Prime Minister says this, and the health secretary is saying that and they’re just contradicting themselves. So that confuses people.”*
(P03, mixed race African/Caribbean/White, 61 years, female). 

Lack of clarity about the information on vaccine benefits and safety was also identified as a problem. Participants expressed concerns that COVID-19 vaccination could make people, particularly those with underlying health conditions, seriously unwell. This made weighing up the benefits and risks more complicated.


*“The problem with me is that they said that, if you’ve got underlying health conditions, take the vaccine, it’s going to protect you. Then, when you hear a story about people dying of the vaccine—and they said no, they took the vaccine but they had underlying health conditions, that’s why they died. But surely the aim of the vaccine is to protect them so they don’t die, and if they’re going to die they might as well not have the vaccine you know? So I don’t know. It’s complicated.”*
(P07, Pakistani, 54 years, female). 

The complexity of decision-making was also described by professionals providing information on vaccination. Professionals described their experience of meeting people who analysed the risks and benefits of vaccination and did not see the benefit of vaccination if they were still able to catch the virus and experience symptoms.


*“Often where people don’t follow—I talk a lot on webinars to people, individuals—and where they stop to basically listen is when you say, “Well, you’re vaccinating to protect yourself from severe illness and death”, they can follow that. But if you then say, “Vaccination doesn’t mean that you can’t harbour the virus and that you may still be experiencing symptoms”, that’s when people, I think, make their own internal risk benefit analysis. And they come to the conclusion that that’s not a good enough reason.”*
(P14, Consultant in Public Health). 

#### 3.2.2. Discrimination and Barriers to Healthcare

Professionals described their experiences when participating in community outreach events, answering questions about COVID-19 vaccination from members of the public. One professional talked about a community event with young Black people and their families in east London. Attendees questioned why health professionals were holding events to talk to them about COVID-19 vaccination when similar events and communication had not taken place on other issues, suggesting that these communities themselves feel underserved by health services.


*“So I did quite a few webinars and actually did a face-to-face event with young Black people in August 2021, and that was, again, really interesting. Young people with their parents came in and there was quite a lot of anger around, ‘you’re coming to talk to us about this vaccine’—first of all, just concerns about the vaccine and actually that it was going to be harming people. But actually more than that, someone stood up and said, “Look, you’ve never bothered about anything else. Why are you coming to talk to us about the vaccines?” And I think that’s why I got involved in that. I thought it was a good opportunity that if we can really reach into underserved communities for this, then there should be no reason why we can’t do the same for other health conditions, whether it’s long-term conditions or childhood immunisations, etc.”*
(P16, GP and Clinical Advisor to NHS England). 

Members of the public also called for more action to improve overall health instead of focusing on vaccination specifically and used examples of long-term conditions disproportionately affecting ethnic minority communities, including diabetes.


*“I think health needs to be helped, especially in ethnic minorities, especially I think obesity is a huge problem with ethnic minorities and diabetes obviously is, and that’s kind of related. It’s not just COVID, it’s other things as well which can affect you in the lon- term with these diseases. So I think just improving your overall health instead of just saying, “Just have the vaccine and everything will be better”, because it won’t.”*
(P09, Indian, 42 years, male). 

Professionals reported hearing members of the public express concerns about racism in vaccine development and deployment. One concern was that vaccines were quickly deployed in order to be tested on people of African descent.


*“You know some people said things like “are you aware that they’ve developed it so fast because they want to test it on people of African descent first.””*
(P21, chief executive of a VCSO supporting Black communities).

Another professional described how people were reluctant to share their ethnicity because they were worried that they would be given a different, perhaps less safe, vaccine to people from a White background.


*“I had one woman early on, say, “Why did you ask my ethnicity?” and I said, “So we can make sure it’s equally distributed.” And she looked at me and I was like, “Do you think there’s like a White cupboard and a Black cupboard?” She was like, “Yeah, basically”, and told me a story of a friend of hers, who was also Black, who was a carer, who had a person she was caring who was White. And so she had her vaccine because it came out of the same vial. So I think we have to be really respectful now, in particular, of people’s decisions.”*
(P17, Head of Community Public Health for a Local Authority). 

Professionals reflected on the community engagement work that occurred during the vaccine campaign, including what had gone well and how this could be developed in future public health campaigns. Professionals suggested putting the same amount of resources into addressing other health problems that are commonly seen in ethnic minority communities, including diabetes and mental health.


*“Other things that I think worked well outside when it came to the uptake of the vaccines, especially in ethnic minority communities, was a recognition of the historical injustice and this idea of trust. I think trust was a big issue. I found that in spaces where there was good uptake, when there was good engagement, there were places where professionals and practitioners acknowledged the historical wrongs and were willing to kind of say, ‘you know what, I understand why you do not trust us. We understand that you were wronged, we are partners on this journey together, I would like you to….’ I think that actually did work well, especially you know, where communities were hesitant.*



*And I find that that is a model that we would have to adopt moving forward. I think we’ve done some good work, some fantastic work in trying to engage with communities, in trying to gain trust around COVID. I think it will be terrible and incredibly dangerous if we do not apply the same amount of energy, energy and drive to diabetes to mental health, to prostate cancer etc.”*
(P18, Public Health programme Lead for a Local Authority). 

#### 3.2.3. Personal Beliefs, Characteristics and Experiences Influencing Vaccine Choice

Religious beliefs, age and occupation emerged as personal factors that participants considered when making their vaccine choices. Participants talked about how their religious beliefs influenced their values around taking the COVID-19 vaccine. One participant described whether he becomes unwell from COVID-19 or not is determined by God rather than vaccination.


*“Some people think, “Oh, if I’m going to die of COVID, I might as well take the vaccine and get less side-effects from it, from the severe form of the virus, and then at least I won’t die.” But whether you’re going to die or you’re not going to die, as a Muslim we leave that in the hands of God. So God decides who’s going to die and who’s going to live. So you know, it’s not in our hands. So you could take the double-jab, get COVID, get it severe and pass away. It can, it can happen. It’s happened to family members.”*
(P07, Pakistani, 54 years, female). 

Four (27%) members of the public who were interviewed worked in health and social care. They talked about their professional experiences during the pandemic, including inadequate equipment, exposure to infection due to insufficient personal protective equipment (PPE), and consultation on mandatory vaccination (which was later rejected). Participants expressed views against mandatory vaccination due to the policy disregarding healthcare workers’ personal, political and religious views.


*“The problem with mandatory vaccination is that from my personal view, you’re asking somebody to say we’re going to force you to put a chemical into your body against your will. And I think that’s how it should be presented, because ultimately that is what we’re asking people to do if we’re saying it’s mandatory. We’re saying you have to put this chemical into your body, regardless of your opinion, whether it’s personal or political or religious or spiritual, whatever it is. And I just think that’s completely wrong to do that.”*
(P09, Indian, 42 years, male).

Participants described how mandatory vaccination forced healthcare workers to consider how much they wanted to stay in their roles in the NHS compared to how strongly they felt about being vaccinated.


*“For me, it’s not about information, it’s about how comfortable you are about putting something in your body. It’s just a question of are you comfortable or not, and at this point, I’m thinking about it but I’m actually thinking to myself that actually I’m ready to leave the NHS [laughs] rather than take it. Because it’s that serious where they’re giving you this ultimatum that you take it or leave your job.”*
(P10, Black African, 43 years, female).

#### 3.2.4. Social Networks and Community Cohesion

Participants described the roles of social media, family, friends, neighbours, community organisations and Community Champions in sharing vaccine information and contributing to vaccination choices.

Participants discussed the role of social media as a source of information in ethnic minority communities, how and why it was used, and the impact it had on decision-making about vaccination. Professionals expressed concerns that some people developed mistrust for the government and NHS due to having limited sources of information apart from word-of-mouth and social media. They also described the use of social media in spreading misinformation or information discrediting scientific findings.


*“A lot of the webinars that I’ve been on, you can sense those undertones of mistrust in the Government and the NHS, exacerbated by a restricted channel of information and education that people have. Which means that people are more reliant on word-of-mouth and also, social media—WhatsApp videos, I think, has a lot to answer for. I think they’ve allowed fringe ideas to become very mainstreamed and spread very quickly. So that’s kind of what we were up against and a lot of that came out at the webinars that I’ve done since then, these myths that started right at the start.*



*And some of them are actually not necessarily myths. Some things are based on a couple of scientific reports that then got blown out of proportion. And even though they were disproven after a while, the damage was already done.”*
(P16, GP and Clinical Advisor to NHS England). 

Members of the public also talked about relying on online information from social media due to it being easier to access, particularly during lockdown.


*“When people are at home, because you cannot go out, the only source they get information from is the internet and social media and news, you know? Because you will only believe whatever you see on those. Like, you can access information from here.”*
(P06, Bangladeshi, 32 years, male). 

Members of the public talked about the way social media has replaced traditional media, such as newspapers, as a source of information and is perceived as a reliable source of news.


*“You look at Instagram just like a new generation of the newspaper, so you look at it as your daily news.”*
(P08, Bangladeshi, 21 years, male). 

Leaders of community organisations supporting ethnic minority communities talked about their approaches to discussing COVID-19 vaccination with the public. They described the need for respect for people’s age, beliefs, historical relationships with medications and concerns about vaccination.


*“There was a respect in terms of age, there was a respect in terms of people’s belief system. Because what we recognised was that we had to do things in the context of people’s historical relationship with medicines and so forth, particularly when they were designed at such rapid speed. So people’s suspicions, we had to roll with that resistance. And what happened is the wonderful thing about being culturally informed and culturally aware was that we gained much more rich information about the individual because they were comfortable in conversing with us.”*
(P21, chief executive of a VCSO supporting Black communities). 

Participants described family, friends and neighbours as their immediate contacts in their social networks. One participant talked about her neighbours in a block of flats in Tower Hamlets and their sense of collective feeling about not being vaccinated.


*“There’s about three or four of us, sometimes. We’re like little old ladies, standing on the landing, talking about all sorts of issues, and the COVID and the vaccination and, you know? And I think, out of us, only one’s had it. You know? And the rest of them has the similar feeling that I’ve got, you know what I mean? We just talk about the same thing and, the way I say I feel, that’s them, as well.”*
(P12, Black Caribbean, 67 years, female). 

Professionals who reflected on successes from community engagement programmes described the role of Community Champions in providing tailored, simple information to the public in formats and channels that were known to be easier to access, including WhatsApp.


*“Champions programmes are where public health teams or the NHS (the statutory body) provides tailored, simple information to people in a form that is easy for them to digest and share…And in Newham, we’ve used WhatsApp as well as email and Zoom. I live in [London borough] and I’m a champion, and we get our information by email. And to be honest with you, I don’t share it, because it’s a lot of words and I have to cut and paste it into WhatsApp and I don’t. Whereas in Newham, we did one page infographics as JPEGs that people could send on, and we were told that they were really powerful. And the biggest lessons from that are—these are really like pedantic—but make stuff that’s easy for people to share. So put stuff in JPEG, not PDF, send it on WhatsApp, not email, and do it quickly when the information changes so that people trust it and they’re able to use it.”*
(P17, head of community public health for a local authority). 

### 3.3. Second-Level Analysis

Overarching themes from the first-level analysis (lack of information credibility and clarity; discrimination and barriers to healthcare; personal beliefs, characteristics and experiences influencing vaccine choice; social networks and community cohesion) were mapped onto Stones’ structuration theory [15,18,19]. Themes were grouped as either ‘external’ or ‘internal’, depending on whether they represent the external structural context in which people operate or whether they represent the internal processes (morals, principles, attitudes and knowledge) that people use to make COVID-19 vaccination decisions. This is represented in the conceptual framework in Figure 1.

## 4. Discussion

### 4.1. Key Findings

This was a qualitative study aiming to improve the understanding of COVID-19 vaccine confidence and the structural factors influencing confidence in the low uptake of ethnic minority groups in London and surrounding areas. The reasons for low uptake are complex and multifactorial. We identified four overarching themes that influenced COVID-19 vaccine confidence: (i) lack of information credibility and clarity; (ii) discrimination and barriers to healthcare; (iii) personal beliefs, characteristics and experiences; and (iv) social networks and community cohesion.

Our findings on information credibility and clarity are consistent with the existing literature on ethnicity data and public trust. In the UK, ethnicity data relating to the number of COVID-19 deaths and vaccinations were recorded using NHS healthcare records and self-reported to health professionals. Researchers have highlighted the poor quality of ethnicity data collection in the UK [43]. Reasons for this have been suggested as a lack of clarity about why ethnicity data is collected, concerns about racial discrimination and a belief that ethnic categories are not representative [43,44,45]. Our findings on public trust in official information and governemnt are consistent with evidence from studies in the UK and US, including co-produced research with UK ethnic minority communities, which suggests that trust in organisations and individuals promoting COVID-19 vaccines is amongst the most important factors in determining uptake [46].

Discrimination was a driver for the lack of vaccine confidence due to the neglect of ethnic minority communities from health services (particularly in the prevention and treatment of health conditions that are prevalent in ethnic minority communities), the impact of structural racism, and concerns about racism in vaccine development and deployment. There is emerging evidence on the role of racism and discrimination as a driver of ethnic health inequalities [47,48,49], mediated through structured social and economic inequalities, including in health behaviours, psychosocial stress and differential access to material, social and healthcare resources [48]. We also identified concerns about racism in vaccine development and deployment. This is consistent with research exploring the views of ethnic minorities and vulnerable communities towards their participation in COVID-19 trials, which found that African and African-Caribbean participants feared vaccines were developed to eradicate Black people [50]. Research with community organisations since the start of the COVID-19 vaccine programme has identified mistrust in vaccines due to previous unethical research (including the US Tuskegee syphilis study) [51]. Asian, Black and Mixed ethnic groups are under-represented or incorrectly classified in UK COVID-19 clinical trials [52], which is likely to further perpetuate these concerns.

Religious beliefs, age and occupation emerged as personal characteristics that influenced vaccine confidence, with some people believing that they could not protect themselves from COVID-19 through vaccination as it is “in the hands of God”. Cultural and religious influences have also been identified in previous studies, which have suggested that working with mosques and faith leaders may be effective in addressing vaccine confidence [53]. Participants working in the NHS described being less willing to be vaccinated following the consultation on mandatory vaccination for health and social care staff and having a preference for losing their jobs rather than being forced to be vaccinated. Previous studies have suggested that ethnic minority healthcare workers are less likely to be vaccinated against COVID-19 than their White colleagues [54], with reasons that include fear around institutional pressure to be vaccinated [7].

Participants described the roles of social media, family, friends, neighbours, community organisations and Community Champions as drivers of vaccine confidence. People had access to vaccine misinformation through social media but also relied on it as a way of easily accessing up-to-date news and information. This is consistent with a 2020 study of sources of information across diverse ethnic groups in the UK, which found that family and social media were sources of information for some people and that negative WhatsApp messages spread rapidly among social networks [53].

Representatives of community organisations described taking a listening approach as a successful way of engaging with the public, as well as respecting existing beliefs and values. Community Champions networks were identified as a successful way of sharing reliable vaccine information and improving vaccine confidence with local communities, particularly when shared in formats that were easily accessible. This is consistent with a study of barriers and facilitators to COVID-19 vaccine uptake in ethnic minorities in primary care, which found that channelling information through local community organisations and Community Champions networks was a key factor in influencing uptake [55].

The theory of structuration has previously been used in research on food security [56,57] to inform our understanding of how systemic forces determine why Black and other ethnic minority groups are disproportionately at risk of food insecurity and other health inequities. Although other studies have highlighted the role of structural racism in reducing vaccine confidence in ethnic minority communities [58,59], to our knowledge, this is the first study to use structuration to explore vaccine inequalities in ethnic minority communities.

### 4.2. Strengths and Limitations

The strengths of this study are the qualitative exploration of experiences and views and the diversity of the participants. Both members of the public and professionals were interviewed, and members of the public all self-identified as either currently or previously refusing vaccination (35% are unvaccinated). This provided insights into both the lived experience of having low COVID-19 vaccine confidence and the experience of professionals providing vaccine information and supporting communities with access to vaccination. The professionals represented a range of organisations involved in the vaccine rollout, including the NHS, public health, local authorities and community groups. We were able to interview people from ethnic minority backgrounds with the lowest vaccine uptake (African, Caribbean, Bangladeshi and Pakistani) and collect demographic data on the country of birth, first language, vaccination status, education level and occupation. A total of 27% of the members of the public worked as healthcare professionals, which revealed insights into the experience of working in the NHS during the COVID-19 pandemic and views towards mandatory vaccination.

PPI involvement was a further strength, particularly the preparatory community engagement work [31], which supported PPI involvement. A data validation workshop with participants and PPI ensured that the analysis took a participatory approach and that the findings were grounded in lived experiences.

Limitations of the study include the focus on one geographic area (London and the southeast) and the number of participants (*n* = 14 members of the public and *n* = 8 professionals). Although London is an area of high diversity and low COVID-19 vaccine uptake, this limits the generalisability of the findings. The number of participants within each ethnic group was small, limiting our ability to explore between-group differences in more depth and compare the views of different groups. This research has demonstrated that vaccine confidence is highly sensitive to social context and perceived risk, and the timing of the research (December 2021–March 2022) may have influenced participants’ responses. Although interpreters were available, all interviews were conducted in English, excluding people who may have had different experiences and views of COVID-19 vaccination due to language. All interviews were conducted by SP, who is a female academic GP of Indian ethnicity. SP’s role as a healthcare professional could have reduced trust or resulted in social desirability bias.

### 4.3. Implications for Research, Policy and Practice

Participant views reflect the strengths of community organisations and Community Champions networks in supporting people from ethnic minority communities. However, there are several concerns influencing the decision to be vaccinated that are not addressed by these strategies. Barriers mediated by social structures cannot be addressed through community-led information and engagement alone.

Recommendations, therefore, address social factors that contribute to ethnic inequalities. The thematic framework based on structuration theory has been used to develop recommendations for policy-makers and healthcare professionals to improve confidence in COVID-19 vaccination in ethnic minority groups (see Figure 2).

With regard to the ethnicity data, improved collection and reporting processes are needed, including working with communities to design appropriate ethnicity categories and explaining the rationale for data collection. Neglect of ethnic minority communities by health services needs to be rectified by creating a more open dialogue about vaccination to allow an exploration of concerns and tailored advice prior to vaccine offers. More funding and resources need to be allocated to clinical conditions disproportionately affecting ethnic minority communities or where ethnic minority communities experience poorer outcomes. In terms of vaccine development, greater transparency is needed in the vaccine development process through improved public engagement and involvement in research. Greater representation by ethnic minority communities in clinical trials could be achieved if regulations from research funders and culturally adapted recruitment processes are adopted. Another research recommendation is for more interdisciplinary research combining both health and sociological perspectives to understand structural inequalities (including the impacts of socioeconomic position and experience of racism and discrimination). To support the ethnic minority workforce in the NHS, workplace policies and practices are needed that support and protect staff wellbeing.

## 5. Conclusions

It may be possible to improve confidence in COVID-19 vaccination for some people from ethnic minority communities by using culturally appropriate and accessible information via community organisations and Community Champions. However, the accumulated morals, views and attitudes and the external structures that shape them (including institutional infrastructures and power relations, political and medical ‘authority’, social position, identity and interactions, and race, racial inequality and racialized social environments) are less amenable to change through information-sharing alone. More research is needed to explore the impacts of external social structures on ethnic health inequalities and actions to address these and achieve ethnic health equity.

## Figures and Tables

**Figure 1 ijerph-20-07008-f001:**
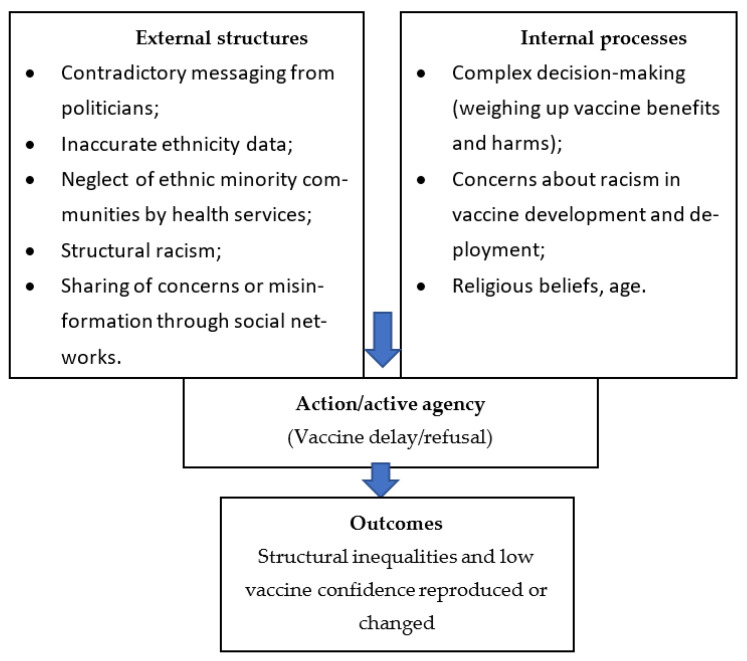
Thematic framework of factors influencing low COVID-19 vaccine confidence in ethnic minority communities (adapted from Greenhalgh and Stones [19]).

**Figure 2 ijerph-20-07008-f002:**
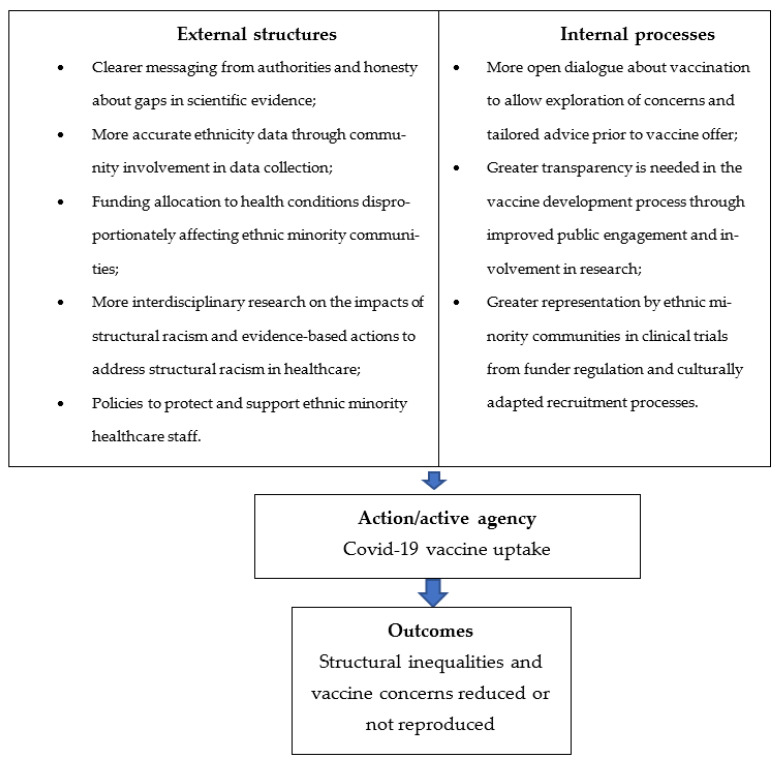
Thematic framework of recommendations for increasing COVID-19 vaccine confidence in ethnic minority communities (adapted from Greenhalgh and Stones [19]).

**Table 1 ijerph-20-07008-t001:** Demographic characteristics of members of the public.

Variable	Mean (SD) or Frequency (%)
Age (years)	45.3 (12.9)
Gender	
Male	5 (35.7)
Female	9 (64.3)
Ethnicity	
Bangladeshi	4 (28.6)
Caribbean	3 (21.4)
Pakistani	2 (14.3)
Mixed	2 (14.3)
African	1 (7.1)
Indian	1 (7.1)
Prefer not to say	1 (7.1)
Country of birth	
UK	7 (50)
Pakistan	2 (14.3)
Bangladesh	1 (7.1)
France	1 (7.1)
St Kitts	1 (7.1)
First Language	
English	9 (64.3)
Bengali	3 (21.4)
French	1 (7.1)
Punjabi	1 (7.1)
Highest level qualification	
No formal qualification	0
GCSE or equivalent	3 (21.4)
Apprenticeship	0
A Level or equivalent	2 (14.3)
Bachelor’s degree or postgraduate qualification	7 (50)
Other qualifications of unknown level	2 (14.3)
Vaccine status	
Unvaccinated	5 (35.7)
One dose	1 (7.1)
Two doses	4 (28.6)
Fully Vaccinated	2 (14.3)
Prefer not to say	2 (14.3)

## Data Availability

Data is contained within the article.

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
