# Peer review of "Beyond Information Provision: Analysis of the Roles of Structure and Agency in COVID-19 Vaccine Confidence in Ethnic Minority Communities"

_ijerph, 2023, doi:10.3390/ijerph20217008_

Round 1

Reviewer 1 Report

Comments and Suggestions for Authors

Overall the paper is good but sometime it gives impression of using lengthy. If possible reduce the extra details. For example

“People from Black and Asian backgrounds are more likely to die from Covid-19 but less 20 likely to be vaccinated, threatening to exacerbate health inequalities already experienced by ethnic 21 minority groups.”

Abstract and intro start with the same sentence……don’t write the same sentences.

Topic of the paper also needs to be revised must be short

Line 538-41 rewrite more clearly “Researchers have highlighted inconsistencies and poor quality of ethnicity data collection (40) due to people feeling less 539 willing to share their ethnicity due to lack of explanation about why the data is being collected, concerns about racial discrimination and a feeling of ethnic categories not being representative (40-42)”

References are not in line with journal’s requirement.

I feel thematic framework’s repetition is not essential it may be avoided.

Strength limitation section is lengthy. Reduce it

I suggest to accept the paper after minor revisions.

Comments on the Quality of English Language

Overall the paper is good but sometime it gives impression of using lengthy. If possible reduce the extra details. For example

“People from Black and Asian backgrounds are more likely to die from Covid-19 but less 20 likely to be vaccinated, threatening to exacerbate health inequalities already experienced by ethnic 21 minority groups.”

Abstract and intro start with the same sentence……don’t write the same sentences.

Topic of the paper also needs to be revised must be short

Line 538-41 rewrite more clearly “Researchers have highlighted inconsistencies and poor quality of ethnicity data collection (40) due to people feeling less 539 willing to share their ethnicity due to lack of explanation about why the data is being collected, concerns about racial discrimination and a feeling of ethnic categories not being representative (40-42)”

References are not in line with journal’s requirement.

I feel thematic framework’s repetition is not essential it may be avoided.

Strength limitation section is lengthy. Reduce it

I suggest to accept the paper after minor revisions.

Author Response

Response to Reviewer 1 Comments

Point 1: Overall the paper is good but sometime it gives impression of using lengthy. If possible reduce the extra details. For example “People from Black and Asian backgrounds are more likely to die from Covid-19 but less 20 likely to be vaccinated, threatening to exacerbate health inequalities already experienced by ethnic 21 minority groups.”

Response 1: Thank you Reviewer 1 for the review and suggestions. I have removed this sentence from the Introduction but kept it in the Abstract as I feel this is an important point to make.

Point 2: Abstract and intro start with the same sentence……don’t write the same sentences.

Response 2: As per reponse 1.

Point 3: Topic of the paper also needs to be revised must be short

Response 3: It’s unclear as to whether the reviewer means ‘topic’ or title’ here. I cannot change the topic of the paper but have suggested shortening the title to “Beyond information provision: a critical analysis of the role of structure and agency in Covid-19 vaccination in ethnic minority communities”.

Point 4: Line 538-41 rewrite more clearly “Researchers have highlighted inconsistencies and poor quality of ethnicity data collection (40) due to people feeling less 539 willing to share their ethnicity due to lack of explanation about why the data is being collected, concerns about racial discrimination and a feeling of ethnic categories not being representative (40-42)”

Response 4: I have tried to clarify this sentence by re-writing it as follows: “Researchers have highlighted the poor quality of ethnicity data collection (40) in the UK. Reasons for this have been suggested as lack of clarity about why ethnicity data is collected, concerns about racial discrimination and a belief that ethnic categories are not representative (40-42).”

Point 5: References are not in line with journal’s requirement.

Response 5: The references have been prepared in line with the journal’s requirements here: https://www.mdpi.com/journal/ijerph/instructions#preparation. References have been numbered in order of appearance in the text (including table captions and figure legends) and listed individually at the end of the manuscript. References have been prepared with the bibliography software package Endnote. References include the full title and are listed as follows: 1. Author 1, A.B.; Author 2, C.D. Title of the article. Abbreviated Journal Name Year, Volume, page range. I have updated the brackets so that they are now square rather than round.

Point 6: I feel thematic framework’s repetition is not essential it may be avoided.

Response 6: It is not clear where in the manuscript the reviewer is referring to (no line number given). If this is referring to the use of Figure 2 as a visual representation of the framework for recommendations, the authors feel this helps to clarify the recommendations and how they relate to structuration theory. As this is a figure rather than text it doesn’t repeat the text but presents it in a different way. 

Point 7: Strength limitation section is lengthy. Reduce it

Response 7: The Strengths and Limitations section has been reduced by removing lines 625-629.

Reviewer 2 Report

Comments and Suggestions for Authors

very low level. in my opinion this should not be allowed for review. No analysis of the literature, invented assumption, no confirmation of the assumed state of affairs, no research questions of the hypotheses. There are no statistics to support the assumptions.

Author Response

Response to Reviewer 2 Comments

Point 1: very low level. in my opinion this should not be allowed for review. No analysis of the literature

Response 1: In line with journal requirements, an analysis of the literature is included in the Introduction and Discussion sections. In the Introduction (lines 47-56) the literature describing the context, purpose and signficance of the research is included (specifically inequalities in Covid-19 vaccine uptake amongst ethnic groups). In lines 59-63 the literature on vaccine hesitancy is included and key publications cited. In lines 64-71, the state of the research field (vaccine hesitancy in ethnic minority communities) is summarised. And in lines 71-77 the state of the research field on the impact of structural factorsand racism on health inequalities is summrised.

In lines 74-103 the literature on the theoretical underpinnings of the study (structuration and social realist theory) are described and key publications cited.

In the discussion (lines 504-609) the findings are interpreted in perspective of previous studies and of the working hypotheses.

Point 2: invented assumption, no confirmation of the assumed state of affairs

Response 2: It is unclear if Reviewer 2 is referring to a particular part of the manuscript as no line reference given. The ‘state of affairs’ in terms of Covid-19 vaccine uptake and Covid-19 vaccine hesitancy in ethnic minority communities is evidenced with peer-reviewed literature from high impact journals in lines 47-77 as described in Response 1. 

Point 3: no research questions of the hypotheses.

Response 3: The main aim of the research is given in lines 105-107 and the objectives are stated in lines 108-114.

Point 4: There are no statistics to support the assumptions.

Response 4: This is a qualitative study therefore only qualitative data (from interviews and focus groups) were collected, analysed and presented.

Reviewer 3 Report

Comments and Suggestions for Authors

This manuscript describes a well-designed qualitative study to uncover drivers of vaccine hesitancy and resistance among minority populations in England (specifically London). The paper is well-written and provides valuable insights for addressing structural racism and developing authentic community engagement practices in order to begin to heal long-standing wounds.

Strengths:

  • As described above.
  • Good use of frameworks to guide design and interpretation or results.
  • The inclusion of professionals provides added context to the work.

Weaknesses:

  • It is unclear whether the professionals were part of the focus groups or not. If so, the authors should discuss the impact this may have had on the comfort level of the public members’ participation.
  • Supplement 1 was not available to the reviewer.
  • The availability of interpreters is to be commended, but it was stated later that the interviews were conducted in English. So were no interpreters used?
  • The number of public and professional participants is inconsistent throughout the document (either 15 and 7, or 14 and 8). See Lines 620, 204, 28, and Table 1 gender.
  • Demographics of the professionals should be included.
  • Part of Figure 1 is cut off.
  • Line 252 is incomplete.
  • Remove Lines 199-201.
  • Line 74 – should this say “unpack” rather than “unpick”?

Author Response

Response to Reviewer 3 Comments

Point 1: It is unclear whether the professionals were part of the focus groups or not. If so, the authors should discuss the impact this may have had on the comfort level of the public members’ participation.

Response 1: Thank you Reviewer 3 for the review and suggestions. I have added a sentence to lines 121-122 clarifying that professionals were not part of the focus groups. Lines 205-207 have also been updated to: “Focus groups and interviews were conducted fourteen members of the public and 8 interviews conducted with professionals involved in Covid-19 vaccination campaigns (a total of N=22).”

Point 2: Supplement 1 was not available to the reviewer.

Response 2: Apologies for this oversight. The topic guide has been uploaded for use as Supplement 1.

Point 3: The availability of interpreters is to be commended, but it was stated later that the interviews were conducted in English. So were no interpreters used?

Response 3: I have added to lines 146-147 that interpreters were offered but not used.

Point 4: The number of public and professional participants is inconsistent throughout the document (either 15 and 7, or 14 and 8). See Lines 620, 204, 28, and Table 1 gender.

Response 4: I have corrected the inconsistences by stating the number of public participants was 14 and the number of professionals was 8 in line 28.

Point 5: Demographics of the professionals should be included.

Response 5: Gender and ethnicity of professionals has been added to lines 207-208. Information on age, country of birth, first language, education and vaccination status was collected from the public but not professionals. This is because the research question explores Covid-19 vaccine decisions made by the public and we were interested in collecting views from a diverse sample of members of the public.

Point 6: Part of Figure 1 is cut off.

Response 6: I have formatted Figure 1 so that the text in the upper two boxes is not cut off.

Point 7: Line 252 is incomplete.

Response 7: Line 252 has been corrected so that it is now complete.  

Point 8: Remove Lines 199-201.

Response 8: Lines removed.

Point 9: Line 74 – should this say “unpack” rather than “unpick”?

Response 9: Unpick updated to unpack in line 74.

Round 2

Reviewer 2 Report

Comments and Suggestions for Authors

dear authors where is this critical analysis?

what method of this analysis did you use?

the abstract has not been corrected yet, it does not contain the purpose, description of the research results. Despite the comments, the authors added only a few words that do not contribute anything, and there are no hypotheses or other things that have been pointed out.

Author Response

Response to Reviewer 2 Comments

Point 1: dear authors where is this critical analysis?

Response 1: The term ‘critical analysis’ comes from critical realist social theory which is described in the ‘Theoretical underpinings’ section in lines 89-109. We have removed the word ‘critical’ from the title and expanded the explanation of critical realist social theory in lines 107-115, with reference to key papers which explain the theory further, as follows:

Within critical realist social theory, the social world is a layered, complex and open system consisting of people (agents) actively and reflexively drawing upon both their attitudes and knowledge (internal structures) and social interactions (external structures) to generate behaviours [22-24]. People belong to and are influenced by multiple institutions and structural relations, which canfacilitate the sharing of common concerns, and make people reflect and act differently [17]. Use of critical realist social theory allowed us to identify the complex interactions between people and social structures and a possible causal pathway for how this may influence Covid-19 vaccine confidence. This is presented in a summary framework.”

References:

  1. Haigh F, Kemp L, Bazeley P, Haigh N. Developing a critical realist informed framework to explain how the human rights and social determinants of health relationship works. BMC Public Health. 2019;19(1):1571.10.1186/s12889-019-7760-7
  2. Scambler G, Scambler S. Theorizing health inequalities: The untapped potential of dialectical critical realism. Social Theory & Health. 2015;13(3):340-54.10.1057/sth.2015.14

Point 2: what method of this analysis did you use?

Response 2: A two-stage analysis of qualitaitve interview data was conducted and is described in lines 190-211. The first stage was an inductive reflective thematic analysis informed by Braun and Clarke. A detailed description is given including the processes of data familiarisation, followed by data coding, validation and grouping into themes. Key papers from the literature on qualitative thematic analysis are referenced.

As described in lines 207-213, in order to go beyond description of the data and produce a more exploratory analysis, we produced a summary of main themes and sub-themes from the first-level thematic analysis, then added to this with an overall thematic framework examining connections between themes and existing structuration and critical realist social theory (illustrated in Table 2 and Figure 1). The practice of producing an overall framework relating findings to theoretical underpinnings is common in qualitative health research and social science research, and is suggested as good practice and explained more fully in this methodology paper from the International Journal of Qualitative Methods:

King (2004) argued that if researchers simply report the codes and themes that appeared in the transcripts, the results will only offer a flat descriptive account with very little depth, doing little justice to the richness of the data. Ideally, as researchers engage in the analytic process, they will progress from description, where the data have simply been organized and summarized to show patterns, to interpretation, where researchers attempt to theorize the significance of the patterns and their broader meanings and implications, often in relation to literature (Braun & Clarke, 2006).”

All references to the literature on qualitaitve ressearch methods are given in the manuscript.

References:

  1. Terry G, Hayfield N, Clarke V, Braun V. Thematic analysis. The SAGE handbook of qualitative research in psychology. 2017;2:17-37,
  2. Pope C. Qualitative research in health care / edited by Catherine Pope, Nicholas Mays. Fourth edition. ed. Pope C, Mays N, editors: Hoboken, New Jersey : Wiley Blackwell; 2020.1-119-41086-X
  3. Canesqui AM, Ritchie J, Lewis J. Qualitative Research Practice: A Guide for Social Science Studentes and Researchers. Botucatu, Brazil: Interface; 2010.
  4. Nowell, L. S., Norris, J. M., White, D. E., & Moules, N. J. (2017). Thematic Analysis: Striving to Meet the Trustworthiness Criteria. International Journal of Qualitative Methods, 16(1). https://doi.org/10.1177/1609406917733847

Point 3: the abstract has not been corrected yet, it does not contain the purpose, description of the research results. Despite the comments, the authors added only a few words that do not contribute anything, and there are no hypotheses or other things that have been pointed out.

Response 3: The purpose of the research is described in lines 26-29. We have updated lines 29-30 to clearly state the aim of the study as follows: “The aim of the study is to draw on health and sociological theories of structure and agency to inform our understanding of how structural factors influence vaccine confidence.”

A description of the the research results is given in lines 34-39. We have updated the text so that the description of the results more closely matches the description given in the Results section and in Figure 1 (Thematic framework of factors influencing low Covid-19 vaccine confidence in ethnic minority communities), as follows: “Our findings suggest that people from ethnic minority backgrounds make decisions regarding Covid-19 vaccination based on a combination of how they experience external social structures (including lack of credibility and clarity from political authority, neglect by health services and structural racism) and internal processes (weighing up Covid-19 vaccine harms and benefits and concerns about vaccine development and deployment).”